# An Advanced Maximum Power Point Tracking Method for Photovoltaic Systems by Using Variable Universe Fuzzy Logic Control Considering Temperature Variability

**Yiwang Wang** [1,2,*], **Yong Yang** [3], **Gang Fang** [4], **Bo Zhang** [2], **Huiqing Wen** [5] , **Houjun Tang** [1], **Li Fu** [6] **and Xiaogao Chen** [7]

1   Department of Electrical Engineering, Shanghai Jiao Tong University, Shanghai 200240, China; hjtang@sjtu.edu.cn
2   School of Electronic and Information Engineering, Suzhou Vocational University, Suzhou 215104, China; zhangbo1221@126.com
3   School of Rail Transportation, Soochow University, Suzhou 215131, China; yangy1981@suda.edu.cn
4   Jiangsu GOODWE Power Supply Technology Co., Ltd., Suzhou 215163, China; kevin.fang@goodwe.com
5   Department of Electrical and Electronic Engineering, Xi'an Jiaotong-Liverpool University, Suzhou 215123, China; Huiqing.Wen@xjtlu.edu.cn
6   Department of New Chemical Materials Engineering, Shandong Polytechnic College, Jining 272067, China; jnfuli@139.com
7   Wuxi Solartale PV Technology Co., Ltd., Wuxi 214174, China; chenxiaogao99@163.com
*   Correspondence: wyiwang@163.com; Tel.: +86-512-6658-6691

**Abstract:** In this study maximum power point tracking (MPPT) is applied to the photovoltaic (PV) system to harvest the maximum power output. The output power of the PV effect changes according to external solar irradiation and ambient temperature conditions. In the existing MPPT strategies, most of them only take variations in radiation level into account, rarely considering the impact of temperature changes. However, the temperature coefficients (TC) play an important role in the PV system, especially in applications where ambient temperature changes are relatively large. In this paper, an MPPT method is presented for a PV system that considers the temperature change by using variable universe fuzzy logic control (VUFLC). By considering the ambient temperature change in PV modules, the proposed control method can regulate the contraction and expansion factor of VUFLC, which eliminates the influence of temperature variability and improves the performance of MPPT, therefore achieving fast and accurate tracking control. The proposed method was evaluated for a PV module under different ambient conditions and its control performance is compared with other MPPT strategies by simulation and experimental results.

**Keywords:** maximum power point tracking (MPPT); photovoltaic (PV) system; variable universe fuzzy logic control (VUFLC); temperature variability

---

## 1. Introduction

With the development of photovoltaic (PV) technologies, an increasing number of PV power generation systems have been presented for large-scale applications. The PV module is one of the key components of PV power generation systems; its performance and efficiency directly affect the high-efficiency operation of the entire system. However, the power energy generated from PV modules relies highly on environmental factors such as solar insolation and the ambient temperature [1,2].

Therefore, in order to harvest the maximum power output and improve the efficiency of the entire PV system, many advanced MPPT control methods have been implemented in PV systems [3–5].

Many MPPT control algorithms have been proposed and developed in recent years [6], such as the classic methods, including open-circuit voltage (OCV)/short-circuit current (SCC) [7,8], incremental conductance (INC) [9], perturbation-and-observation (P&O) [10] and other hybrid strategies [3]. Due to the non-linear problems of PV cells, some soft computing techniques have been applied to the MPPT of PV systems, such as the artificial neural networks method (ANN) [11] and fuzzy logic control (FLC) [12–14].

Most of the MPPT approaches only take the variability in radiation level into account, while rarely considering the effects of temperature. Some new MPPT methods based on temperature measurements were discussed in Reference [15]. Many control algorithms use temperature as a feedback parameter to realize MPPT. For example, studies in References [16–18] proposed an MPPT-temperature algorithm where the PV module temperature was used to determine the maximum power point voltage to track the maximum power point (MPP). In Reference [19], a sun tracking system that included the temperature effect was presented, and an optimum system design was achieved. Compared to other approaches under the same control algorithm, the MPPTs based on temperature measurement directly consider the temperature variations leading to MPP changes, which can obtain a faster tracking speed, especially in engineering applications where temperature changes are relatively large.

Furthermore, to improve the tracking accuracy, some artificial intelligence techniques have been employed for the MPPT implementation [20–22]. Fuzzy logic control (FLC) is a relatively popular and mature artificial intelligence algorithm and has been applied to track the MPP in PV systems [23,24]. In Reference [25], results indicated that FLC had the best performance when compared to some MPPT techniques with INC, P&O, and others in both dynamic response and steady-state under most of the normal operating range. The variable universe fuzzy logic control (VUFLC) can adaptively change the input and output universes to improve the control effect and obtain higher control accuracy [26–29].

However, how to choose a variable universal scalable or contraction-expansion factor is a challenging issue practically, according to the nonlinear characteristic of PV systems [29]. Hence, this works to exploit the PV modules' real-time temperature variable as the constraints of variable universal factor selection, then a new VUFLC-temperature MPPT algorithm was designed to obtain efficient tracking performance in the external working environment (environmental condition) variations. The proposed VUFLC-temperature MPPT method selects the variable universal factor according to the dynamic change of temperature by combining the modules' temperature coefficients (TC) characteristic, which can accelerate the MPPT and improve the tracking accuracy when compared to conventional MPPT strategies. The proposed VUFLC-temperature MPPT method was validated by simulation and experimental results.

This paper is organized as follows: Section 2 describes the characteristics of the PV system. Section 3 presents the proposed VUFLC-temperature MPPT algorithm and its development techniques. Section 4 presents the simulation results. Section 5 provides the results of the experimental tests. Finally, the paper is concluded in Section 6.

## 2. Model and Characteristics of a PV System

PV cells and modules are the key components of PV systems, which absorb photons of light and release electron charges, and can directly convert solar energy into electricity [30]. The electric energy generated from the PV effect is highly dependent on environmental factors [1]. The PV cell is a nonlinear device and can be represented as a current source model [31]. The common and popular PV models are single diode and double diode [4,31,32]. Here, we use the single diode model to describe the characteristics of PV cells.

The single diode equivalent circuit is shown in Figure 1, where $D$ is a parallel diode, $R_{sh}$ is the shunt resistance and $R_s$ is the series resistance. The output mathematical equations were obtained in References [4,30,33,34].

$$I_{pv} = I_{ph} - I_d - I_{R_{sh}} \tag{1}$$

$$I_{pv} = I_{ph} - I_0 \left[ \exp(\frac{q(V_{pv} + R_s I_{pv})}{AKT}) - 1 \right] - \frac{V_{pv} + R_s I_{pv}}{R_{sh}} \tag{2}$$

where $I_{ph}$ is the light-generated current of the elementary PV cell, $I_d$ is the current of the parallel diode, $I_{R_{sh}}$ is the shunt current of resistance $R_{sh}$, and $I_0$ is the reverse saturation current of the diode. $V_{pv}$ and $I_{pv}$ are the output voltage and current, respectively. $q$ is the electron charge ($1.602 \times 10^{-19}$ C), A is the diode ideality factor, and K is the Boltzmann constant ($1.38 \times 10^{-23}$ J/K).

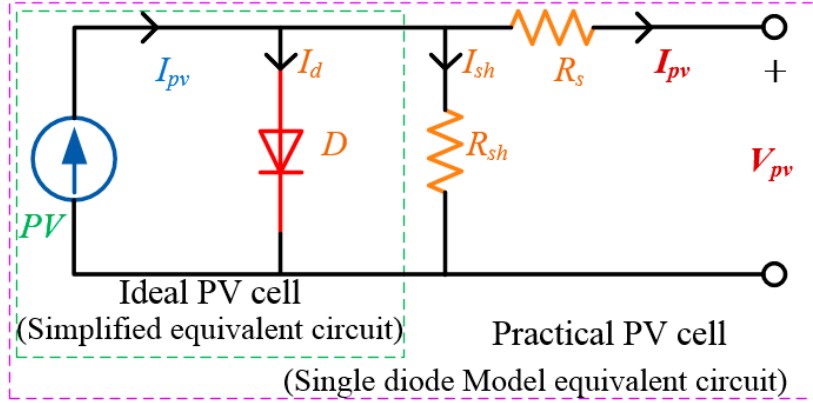

**Figure 1.** Single diode model and equivalent circuit of the photovoltaic (PV) cell.

According to Equations (1) and (2), the electrical characteristics of the PV cell are generally drawn as a current versus voltage (*Ipv-Vpv*) curve and a power versus voltage (*Ppv-Vpv*) curve under different environmental conditions [29].

Figure 2 shows the characteristics of the PV cell in different environmental conditions. As can be seen from the relation between the PV output parameters and the environment variables, it is highly nonlinear and dependent on the solar radiation level and temperature changes on the PV cell [35]. The output power energy of the PV system is affected by radiation and temperature. Figure 2a presents the curves under different radiation, i.e., the current $I_{SC}$ increases quasi-linearly with the radiation while the voltage $V_{OC}$ increases slightly, and the maximum electric power $P_{max}$ changes as the radiation changes. Figure 2b gives the relationship of I_V and P_V at different temperatures, where $I_{SC}$ slightly increases and $V_{OC}$ strongly decreases with temperature changes. The maximum electric power also significantly decreases with a temperature rise, as illustrated in Figure 3 [30].

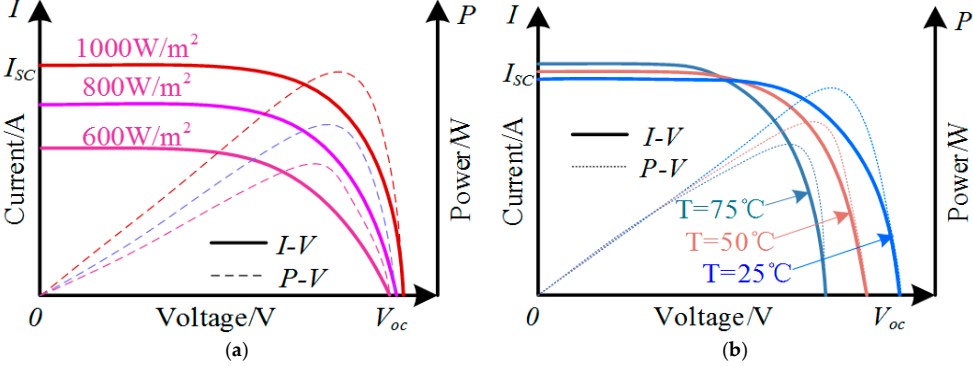

**Figure 2.** Characteristics of the PV cell at different environmental conditions: (**a**) Different solar radiation level effect; (**b**) different temperature variability effect.

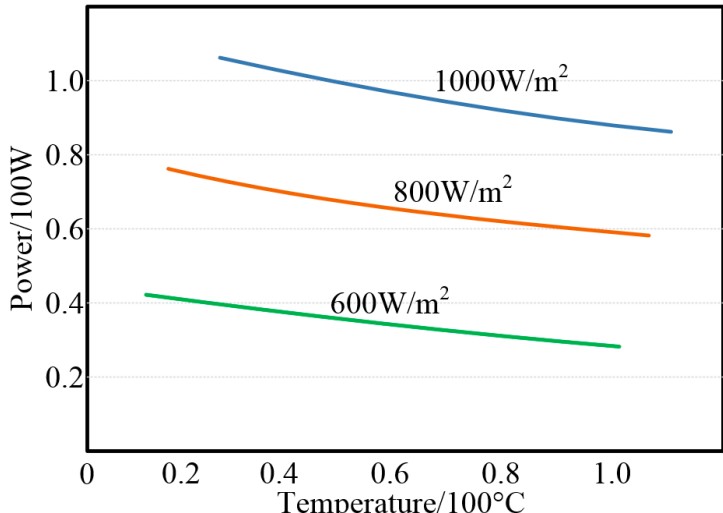

**Figure 3.** Power variations with temperature changes under different radiation levels.

In recent years, many researchers have conducted MPP control studies under radiation conditions and proposed many various MPPT algorithms in the literature [1–30]. This paper focuses on the impact of temperature on MPPT and proposed corresponding control strategies and implementation methods. The short-circuit current $I_{SC}$ and open-circuit voltage $V_{OC}$ at the reference nominal operating cell temperature (NOCT) $T_{NOCT}$ can be calculated at a given temperature TC with some temperature variation [28,30], respectively. The PV output parameters considering temperature effect can be obtained as follows:

$$I_{SC} = I_{SC-NOCT}\cdot[1 + \alpha_{SC}\cdot(T_c - T_{NOCT})] \tag{3}$$

$$V_{OC} = V_{OC-NOCT}\cdot[1 + \beta_{OC}\cdot(T_c - T_{NOCT})] \tag{4}$$

$$P_{max} = P_{max-NOCT}\cdot[1 + \gamma_{max}\cdot(T_c - T_{NOCT})] \tag{5}$$

where the $T_C$ is the operating temperature of the PV cell and $T_{NOCT}$ is the temperature at the nominal ambient environment. $I_{SC-NOCT}$, $V_{OC-NOCT}$, and $P_{max-NOCT}$ are the short-circuit current, open-circuit voltage, and maximum power at the reference NOCT, respectively. $\alpha_{SC}$, $\beta_{OC}$, and $\gamma_{max}$ are the temperature coefficients (TC) of $I_{SC}$, $V_{OC}$, and $P_{max}$, respectively. According to Equations (3)–(5), the short-circuit current, open-circuit voltage and maximum power of PV cells will be affected by temperature changes. Therefore, the impact of temperature variability can be considered when designing the MPPT control strategy through the direct or indirect measurement of the operating temperature of the PV cells.

## 3. MPPT Control System and Proposed Control Method

### 3.1. MPPT Control System

Power electronic converters are commonly applied in a PV system to achieve different MPPT control methods, where the converters act as the interface between the PV source and different loads. In order to efficiently track the MPP, the converter needs to adjust the duty cycle under varying operating atmospheric conditions [1,4,25]. The MPPT controller acquires the real-time operating parameters depending on the control algorithm, then outputs the corresponding control signal to control the DC/DC converter. The most common solar PV MPPT system consists of a PV module, DC/DC boost converter, MPPT controller and a load, as shown in Figure 4. The PV cells generate power energy and its output is connected to the DC/DC converter. The converter is controlled by the MPPT controller where different control algorithms can be carried out.

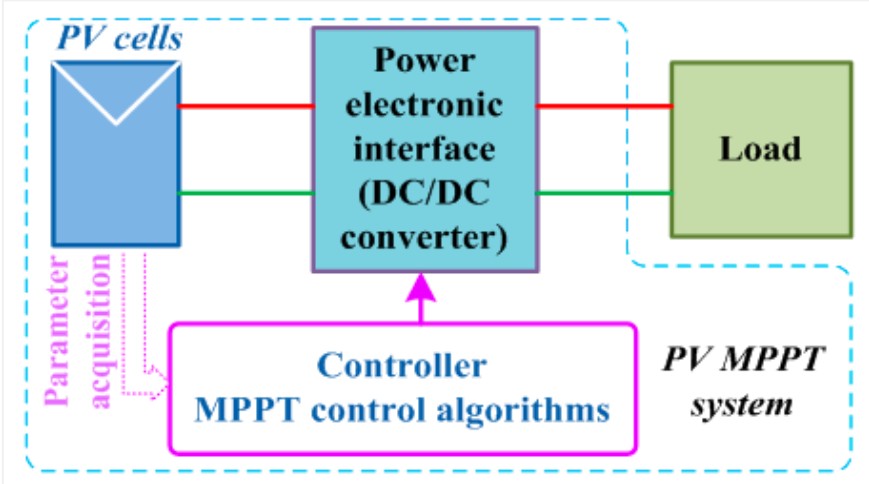

**Figure 4.** The block diagram of a general PV maximum power point tracking (MPPT) system.

*3.2. Variable Universe Fuzzy Logic Control (VUFLC)*

Due to the nonlinear characteristics of the PV system, intelligent MPPT control algorithms in PV systems are very promising and some have been successfully employed for maximum power extraction [4,6]. Fuzzy logic control (FLC) is one of the most prevalent intelligent control techniques, which has advantages like a fast response time, less fluctuation and high control accuracy. Therefore, it is effective in controlling nonlinear systems [4].

However, the conventional FLC with fixed fuzzy control rules will not perform well when working with large uncertainties or unknown variations in the systems [28], and the control precision is commonly not high. Hence, adaptive fuzzy logic controllers (AFLC) have been proposed to solve this issue [26–28]. The VUFLC is one of the AFLC that has been applied to various control engineering projects such as specialty vehicle control [27], analog circuit implementation [28] and liquid lever system [36]. The VUFLC combined with the characteristics and advantages of the variable universe control is introduced into the MPPT control, which can improve the control speed and precision of the PV system.

The VUFLC was proposed in Reference [26] and its discourse universes of the input and output variables can be adjusted according to changed control conditions instead of adjusting the fuzzy rules, thus illustrating more control accuracy and flexibility than conventional FLCs [27,28]. Figure 1 illustrates the process diagram of the variable universe, where Figure 5b shows an original universe with five fuzzy partitions as fuzzy sets of NB (negative big), NS (negative small), ZE (zero), PS (positive small) and PB (positive big) with a piecewise linear membership function [28]. Figure 5a shows the contracting of the universe and Figure 5c presents the expanding of the universe.

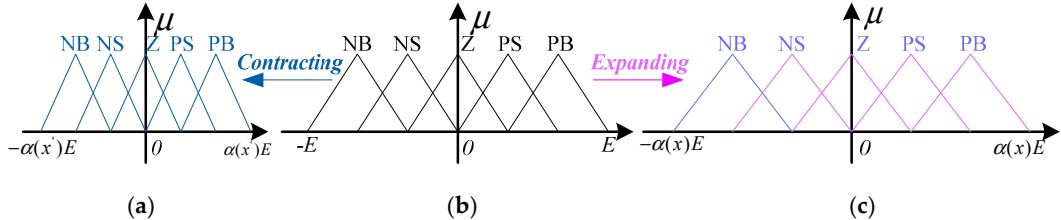

**Figure 5.** Illustration of variable universe: (**a**) Contracted universe; (**b**) original universe; (**c**) expanded universe.

In order to simplify the analysis, common two-input and single-output systems were taken as an example. Let the universes of input $(x_1, x_2)$ and $y$ output variables be $X_i = [-E_i, +E_i]$ ($i = 1, 2$) and

$Y = [-U, +U]$, respectively. According to Figure 5, the input and output universes $X_i$ and $Y$ can be adjusted with the change of variables $x_1$ and $y$, respectively. Their relationship is as follows:

$$X_i(x_i) = [-\alpha_i(x_i)E_i, +\alpha_i(x_i)E_i], i = 1, 2 \tag{6}$$

$$Y(y) = [-\beta(y)U, +\beta(y)U] \tag{7}$$

where $\alpha_i(x_i)$ is the input universe contraction factor and $\beta(y)$ is the output universe contraction factor. With a contraction factor change, the input variable and output variable will change to better adapt to different control conditions and achieve more precise control objectives. Therefore, the design and selection of contraction factors is also important and the details about contraction factors will be introduced based on the proposed MPPT system in the next section.

The fuzzy rule is essential for a VUFLC system, let $A_{x_i}$ and $B_y$ be regarded as linguistic variables of input $x_i$ and output $y$, respectively. The fuzzy IF-THEN control rule [26] is formed as follows:

$$\text{IF } x_1 \text{ is } A_{x_1} \text{ and } x_2 \text{ is } A_{x_2}, \text{THEN } y \text{ is } B_y \tag{8}$$

Unlike the conventional FLC, the universes of VUFLC can correspondingly adjust along with changes to the input variables. The membership of input and output variables use the triangle. For a complete fuzzy controller design, a defuzzification process is needed, that is, the VUFLC output is converted from a linguistic variable to a numerical variable. There are many defuzzification methods, and in this design, the center-of-gravity (COG) defuzzification method was employed [25,28,37].

### 3.3. Proposed Control Method

Figure 6 shows the block diagram of the PV MPPT control system. The ordinary PV module was used for the PV power generation model, the DC/DC was converted and the output was connected to a load.

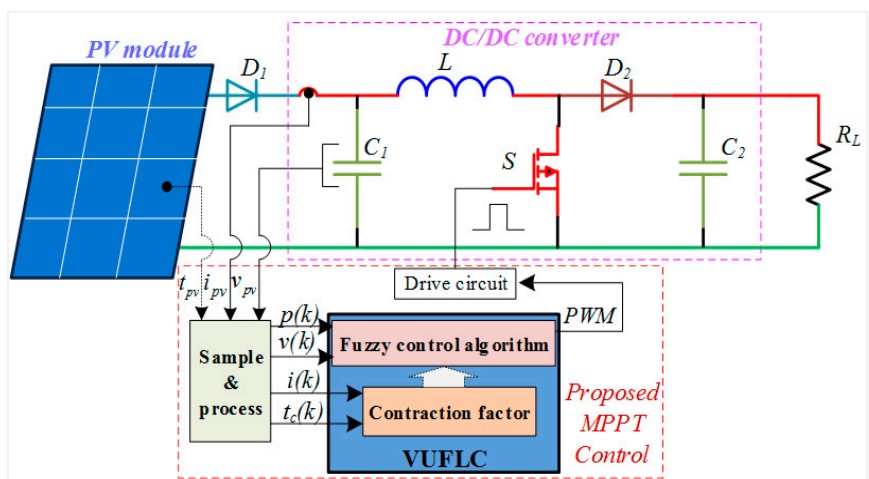

**Figure 6.** Block diagram of the PV MPPT control system.

As previously described, the proposed design steps of the VUFLC applied to PV MPPT system are presented in the remainder of this subsection.

In terms of the input and output variables, the proposed VUFLC has two input variables and one output variable. The two VUFLC input variables are the error $E(k)$ and the difference in error $CE(k)$, which are calculated as follows [38].

$$E(k) = \frac{p(k) - p(k-1)}{v(k) - v(k-1)} \tag{9}$$

$$CE(k) = E(k) - E(k-1) \tag{10}$$

where $k$ refers to the iteration number; $p(k)$ is the instantaneous output power of PV; and $v(k)$ is the instantaneous output voltage of PV corresponding to $k$th sample.

In terms of the fuzzy control rules, according to the actual operation of the PV power generation project and the previous MPPT control engineering experience, the linguistic expressions and the initial universes of the inputs and output variables are given in Table 1.

**Table 1.** The input and output variables and initial universe.

| Parameter | Type | Linguistic | Universe | |
| :---: | :---: | :---: | :---: | :---: |
| | I/O | | Min | Max |
| Power error/Volt error ($x_1$) | Input | NB NS ZE PS PB | −40 | +40 |
| Error change ($x_2$) | Input | NB NS ZE PS PB | −80 | +80 |
| Duty change ($y$) | Output | NB NM NS ZE PS PM PB | −0.09 | +0.09 |

The input and output variables have five and seven linguistic expressions, respectively. The input variables have five linguistic expressions as follows: NB (negative big), NS (negative small), ZE (zero), PS (positive small) and PB (positive big), and the output variables have seven linguistic expressions as follows: NB (negative big), NM (negative medium), NS (negative small), ZE (zero), PS (positive small), PM (positive medium), and PB (positive big), which adds two linguistic expressions. The initial universe of the input variables $x_1$ and $x_2$ are normalized to the range (−40, +40) and (−80, +80), respectively. The initial universe of the output variable $y$ is normalized to the range (−0.09, +0.09). All of the membership functions of the input and output variables use a triangular form, which is easy to calculate and specifies the entire fuzzy partition of these variables.

According to Equations (9) and (10), the sign of the input variable $x_1$ shows if the operating point is located on the left or right side when compared to the actual MPP position P–V curve, while $x_2$ expresses the moving direction of this operation point [38,39]. The output control variable $y$ can be obtained under the fuzzy control rules. Based on Equation (8), the fuzzy control rules are given in Table 2, which determine the VUFLC output control signal.

**Table 2.** The input and output variables and initial universe.

| CE($x_2$) U($y$) E($x_1$) | NB | NS | ZE | PS | PB |
| :---: | :---: | :---: | :---: | :---: | :---: |
| NB | NB | NS | PS | PM | PB |
| NS | NS | PS | PM | PM | PB |
| ZE | NM | NS | ZE | PS | PM |
| PS | NS | ZE | PS | PM | PB |
| PB | NB | NM | NS | PS | PB |

In terms of universe control factor design, the input and output variables can be adaptively adjusted by the contraction factor in Equations (6) and (7), and the variable universe process can be described from Figure 5. The conventional contraction factors are presented and discussed in Reference [26], which achieved the contraction of the universe when the input variable was small. In order to speed up the response time and improve the control accuracy, an improved universe control factor for the input variable was proposed and designed, considering the influence of temperature characteristics. The control factors of the new input variable are defined as follows:

$$\alpha'(x_i) = 1 - \lambda_1 \exp[-k_1(\gamma_{max} \cdot \Delta T_C)^2], 0 < \lambda_1 < 1, k_1 > 0, i = 1, 2; |\Delta T_c| \le \theta_{set} \tag{11}$$

where $0 < \alpha'(x_i) < 1$, and is related to the maximum power temperature coefficient $\gamma_{max}$ and the temperature change value $\Delta T_C = T$. The $\gamma_{max}$ can be found in the manufacturer's datasheet.

In Equation (8), the value of $\gamma_{max} \cdot \Delta T_C$ represents the power change with ambient temperature. When $\Delta T_C$ increases, the $\alpha'(x_i)$ value decreases, the universe of input variables $x_1$ and $x_2$ are expanded, the output power change becomes larger, and VUFCL will achieve fast MPP tracking. However, if $|\Delta T_c| \geq \theta_{set}$ ($\theta_{set}$ is the maximum threshold for the temperature change setting), then $\alpha'(x_i)$ will become 1, and the universe variables take the maximum value. On the other hand, when $\Delta T_C$ drops, the $\alpha'(x_i)$ value rises, the universe of input variables $x_1$ and $x_2$ are contracted, and the VUFLC will limit the oscillations and improve the MPP control tracking accuracy. Similarly, the modified output variable control factor can be expressed as:

$$\beta(y) = 1 - \lambda_2 \exp[-k_2(\beta_{OC} \cdot \Delta T_C)^2], 0 < \lambda_1 < 1, k_1 > 0; |\Delta T_c| \leq \theta_{set} \tag{12}$$

where $\beta(y)$ is related to the $\beta_{OC} \cdot \Delta T_C$, because the output control signal is $\Delta D$, which can adjust and change the voltage ratio of the converter. When the control factor $\beta(y)$ takes the voltage temperature effect into account, the control compensation can be achieved, and the control accuracy is improved.

Different value selection of parameters $\lambda_1$, $\lambda_2$, $k_1$, and $k_2$ will have a certain impact on the range of variable universes and affect the convergence of the proposed control method. Considering the correlation between the variables and actual engineering application, the related parameters were chosen as $\lambda_1 = 0.4$, $\lambda_2 = 0.6$, and $k_1 = k_2 = 10^4$. Hence, $\alpha'(x_i)$ ranges from 0.4 to 1.0, and the range of $E(k)$ and $CE(k)$ are limited to the limits ($-40/-36$, $+36/+40$) and ($-80/-48$, $+48/+80$), respectively. Similarly, the range of U (output universe) is tuned to the limits ($-0.09/-0.054$, $+0.054/+0.09$). Furthermore, the $\theta_{set}$ was selected as 2 °C. Therefore, the proposed VUFLC-temperature can change the input and output universes with control factor variation. According to the selected parameters, the input and output universes control factors $\alpha'(x_i)$ and $\beta(y)$ are calculated by follows:

$$\left. \begin{array}{l} \alpha'(x_i) = 1 - 0.4 \exp[-10^4 \cdot (\gamma_{max} \cdot \Delta T_C)^2] \\ \beta(y) = 1 - 0.6 \exp[-10^4 \cdot (\beta_{OC} \cdot \Delta T_C)^2] \\ else\ \alpha'(x_i) = \beta(y) = 1, |\Delta T_C| > 2\,°\mathrm{C} \end{array} \right\} when |\Delta T_C| \leq 2\,°\mathrm{C} \tag{13}$$

Figure 7 shows the curves of the variable universe control factors.

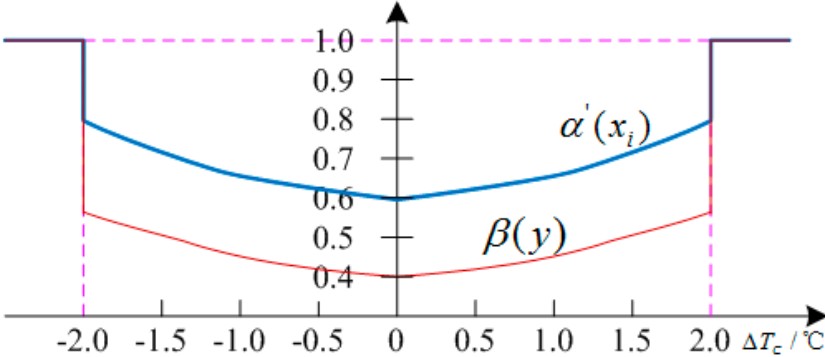

**Figure 7.** Curves of variable universe control factors.

The TC of $P_{max}$, $\gamma_{max}$, $V_{OC}$ and $\beta_{OC}$ were chosen as $-0.40\%/°C$ and $-0.30\%/°C$, which can adjust the universes of the input and output variables to achieve the MPPT of a PV system by using control factors. Therefore, the VUFLC-temperature can improve the control accuracy and reduce power fluctuations in the PV MPPT.

In regard to the implementation of VUFLC for MPP, the proposed VUFLC-temperature MPPT control algorithm was implemented as follows. First, the controller detects the output $V_{pv}$ and $I_{pv}$ of the PV module and computes the $E(k)$ and $CE(k)$, then it measures the PV module temperature $T_C(k)$ and evaluates $\Delta T_C$ (or calculates by related evaluation method of the test standard). According to the different $\Delta T_C$ values, the VUFLC-temperature controller selects different universe control factors

based on Equation (13). Finally, the updated duty cycle control signal is output to control the power converter and achieve MPPT tracking. The detailed and complete control implementation flow chart is shown in Figure 8.

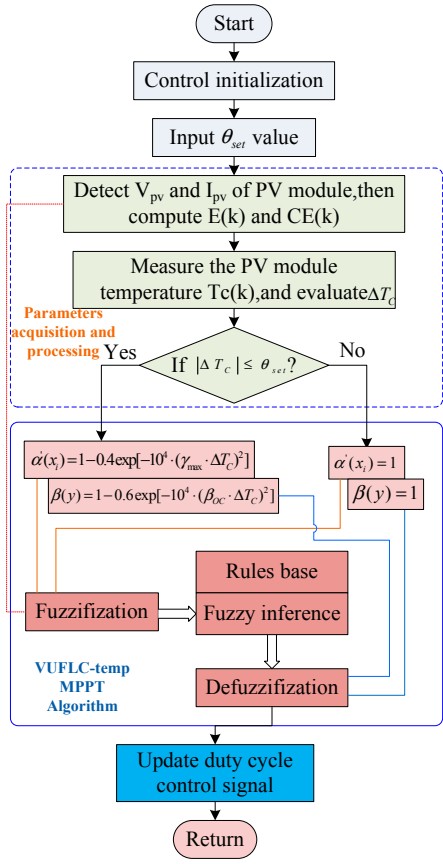

**Figure 8.** Flowchart of the proposed variable universe fuzzy logic control temperature (VUFLC-temperature) MPPT control algorithm procedure.

## 4. Simulation Results

The MPPT control system based on VUFLC-temperature algorithm was simulated and developed to test and confirm the proposed method, as shown in Figure 6. A boost circuit was selected to be the converter to achieve the MPP by adjusting the control signal through the VUFLC-temperature controller. The proposed control algorithm and MPPT system were simulated in MATLAB/Simulink (version 9.1, the MathWorks, Inc., Natick, MA, USA). The simulation model consisted of radiation and temperature input units, a PV module, a converter, a load and the proposed controller. The output of the PV module was connected to the boost converter, then the controller adjusted the duty cycle of the converter control signal to achieve maximum power control. The proposed VUFLC-temperature MPPT PV system is shown in Figure 9. The control logic was implemented through software programming. The PV module, boost converter and load were built with the related components in SimPowerSystems.

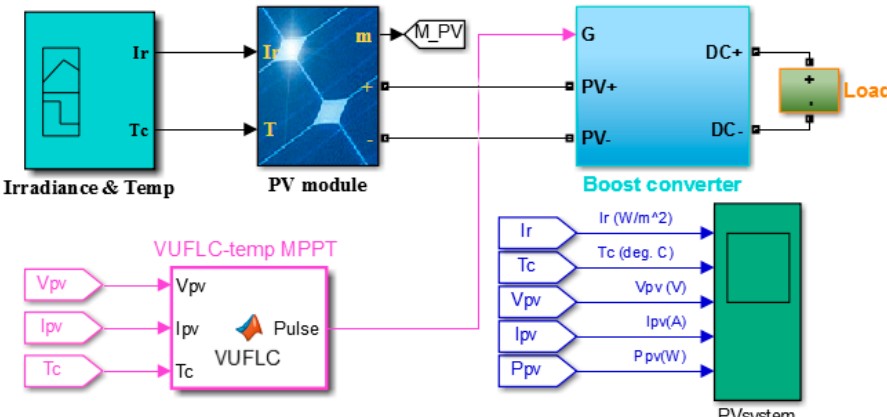

**Figure 9.** Simulation model of the VUFLC-temperature PV MPPT system.

The simulation parameters of the Crystalline Silicon PV module are listed in Table 3. The simulation input was composed of five listed PV modules connected in series and the total input peak power was 1650 W.

**Table 3.** Simulation PV module parameters.

| Electrical(STC) | | Temperature Characteristics | |
| --- | --- | --- | --- |
| Specification | Data | Specification | Data |
| Maximum Power ($P_{max}$) | 330 W | Temperature Coefficient of $P_{max}$ | −0.41%/°C |
| Optimum Operating Voltage ($V_{mp}$) | 37.5 V | Temperature Coefficient of $V_{OC}$ | −0.38%/°C |
| | | Temperature Coefficient of $I_{SC}$ | 0.05%/°C |
| Open Circuit Voltage ($V_{oc}$) | 46.2 V | Nominal Operating Cell Temperature | 45±2 °C |
| | | Operational Temperature | −40~+85 °C |

Figures 2 and 3 demonstrate that the temperature will affect the PV output power and that the MPP of the PV module also shifts with temperature changes [30]. At maximum and minimum operating temperatures per day, the output power can vary by about 20%. The simulation model system (Figure 9) can simulate the temperature and irradiance fluctuation of the input PV array, which indicates the adaptability and superiority of the system under different operation conditions. In order to better demonstrate the proposed control strategy, its control effects were compared with conventional MPPT FLC (fuzzy logic control) and INC (incremental conductance) under the same conditions.

Figure 10 shows that the simulation results of the MPPT control tracking process under the solar radiation intensity remained at 500 W/m$^2$ constantly and the temperature changed slowly; the temperature rose from a minimum of 0 °C to a maximum of 72 °C, then dropped back to the lowest temperature value as seen in Figure 10a, where the PV output voltage and power changed slowly. In Figure 10b, using the proposed method, the output and power could respond quickly with precise tracking. However, in Figure 10c with FLC and Figure 10d with INC, respectively, the output power had relatively large fluctuations at 0.43 s and 0.56 s, which could not accurately adapt to the temperature changes.

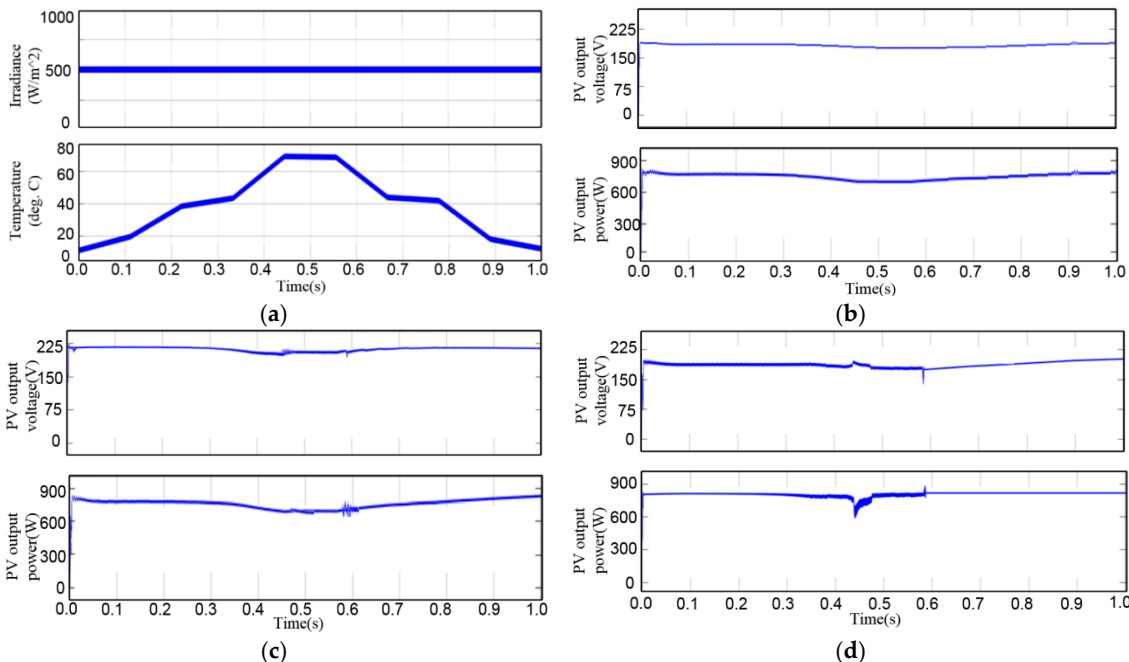

**Figure 10.** MPPT tracking simulation results of different control methods in variable temperature: (**a**) Change temperature and keep the solar radiation constant; (**b**) VUFLC; (**c**) fuzzy logic control (FLC); (**d**) incremental conductance (INC).

In Figure 11, the radiation and temperature simultaneously changed, with random changes during the 1.0 s period. Before 0.3 s, the irradiance and temperature almost rose synchronously, then kept at the maximum of 1000 W/m$^2$ and 60 °C at 0.37 s. At 0.5 s, the radiation declined to 500 W/m$^2$, and at 0.78 s reached 1000 W/m$^2$ again. At 0.715 s, the temperature changed to 25 °C and at 0.92 s back to 60 °C. In Figure 11b, with VUFLC, when the radiation and temperature varied, the power curve had smooth tracking, no power loss and the power ripple maximum power point oscillations were eliminated. In Figure 11c,d, with FLC and INC, the power tracking error and fluctuation were comparatively large, when the temperature changed the power curves overshot and loss were occurred.

Figure 12 shows the simulation results of the MPPT tracking process under the temperature step change. Before 0.4 s, the temperature was 20 °C. At 0.4 s, the temperature step increased from 20 °C to 50 °C. Additionally, before 0.2 s, the initial radiation was 0 W/m$^2$, and it quickly increased to 1000 W/m$^2$ at 0.25 s, then at 0.4 s began to drop to 250 W/m$^2$ at 0.45 s. As shown in Figure 12b, the VUFLC has a quick MPPT response. However, Figure 12c with FLC and Figure 12d with INC had a large overshoot and dynamic error; at 0.4 s, the power tracking with FLC and INC both had power fluctuation loss.

The simulation results for the proposed VUFLC-temperature based MPPT control method are presented and compared to the conventional FLC with temperature change. The variable universe control factors of the VUFLC controller will dynamically adjust according to the change in atmosphere; when the module temperature rises or drops, the input and output universe control factors are chosen with different values to speed up the MPPT and control convergence and all the power tracking waveforms with VUFLC are smooth with less loss and no overshooting, which has a faster tracking speed and more precise control effect than the other methods. It can be seen from the results that VUFLC had a significant impact on the MPP tracking control, where a relatively small universe can improve the control accuracy and reduce the oscillation at the MPP. The comparisons between the simulation results and existing others are briefly summarized in Table 4.

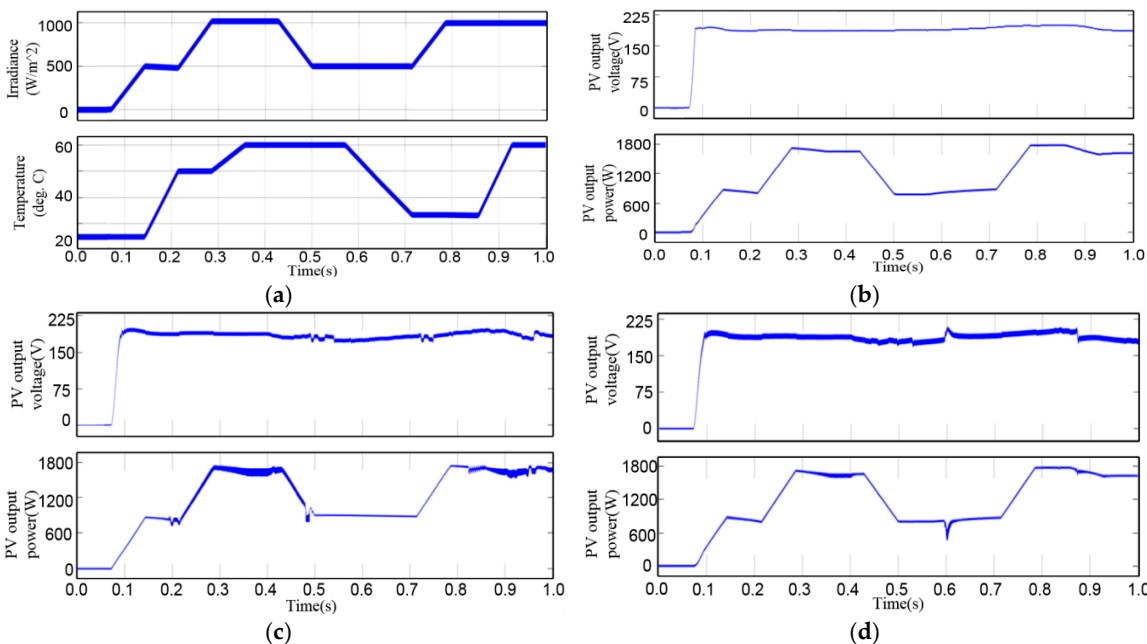

**Figure 11.** MPPT tracking simulation results of different control methods in variable radiation and temperature: (**a**) Changing solar radiation and temperature simultaneously; (**b**) VUFLC; (**c**) FLC; (**d**) INC.

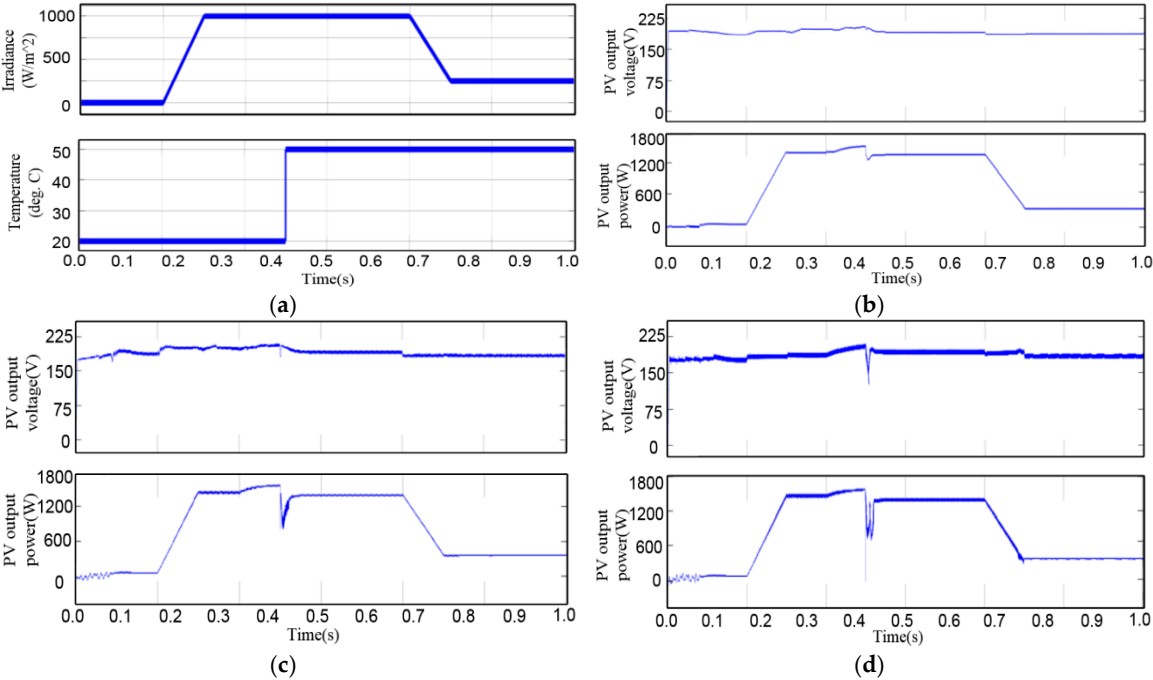

**Figure 12.** MPPT tracking simulation results of different control methods in variable radiation and step temperature: (**a**) Changing solar radiation and step temperature simultaneously; (**b**) VUFLC; (**c**) FLC; (**d**) INC.

According to the simulation and comparative analysis results, the proposed VUFLC-temperature MPPT method had a better control performance, especially under conditions of temperature change. The VUFLC-temperature method could also obtain a fast tracking speed, small oscillation, and improved accuracy during the step temperature change.

**Table 4.** Performance comparison of different MPPT control methods.

| Items | MPPT Methods | | | | |
|---|---|---|---|---|---|
| | **P&O** | **INC** | **ANN** | **FLC** | **Proposed VUFLC** |
| Dynamic response | Poor | Medium | High | Medium | High |
| Transient fluction | Bad | Bad | Good | Good | Good |
| Steady oscillation | Large | Moderate | Zero | Small | Zero |
| Static error | High | High | Low | Low | Low |
| Control accurcy | Low | Accurate | Accurate | Accurate | Excellect |
| Tracking speed | Slow | Slow | Moderate | Fast | Very fast |
| Overall efficiency | Medium | Medium | High | High | High |
| System complexity | Simple | Simple | Medium | Medium | Medium |
| Temperature characteristics | Poor | Poor | Good | Good | Excellect |

## 5. Experimental Validation

In order to further verify the analysis and simulation results, the proposed VUFLC-temperature MPPT control algorithm was experimentally validated on a PV system prototype. A photograph of the experimental prototype hardware is shown in Figure 13.

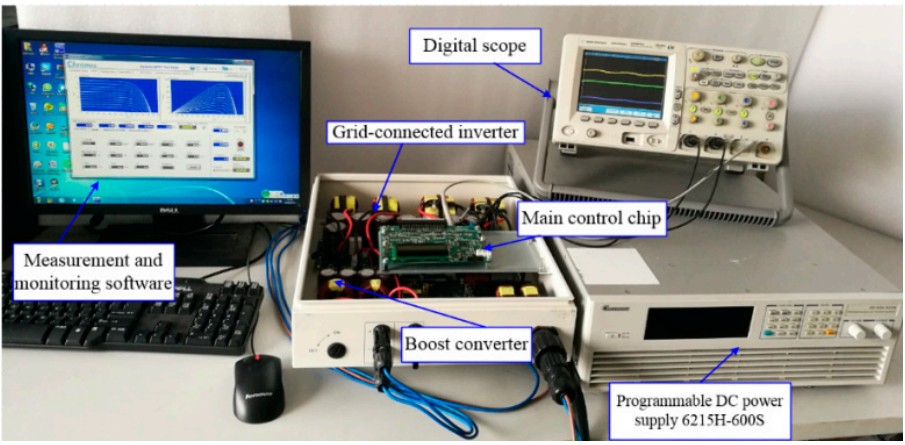

**Figure 13.** Experimental setup of the PV MPPT system.

A PV module analog programmable DC power supply 6215H-600S (CHROMA ATE (SUZHOU) CO., LTD., Suzhou, China) was employed as the input for the test; it was also used to emulate different working environments and temperature changes. The main control chip was a DSP (Digital signal processor) TMS320F28035 which was employed for implementing the proposed control algorithm. A boost converter was employed to achieve the power conversion and MPPT. A grid-connected inverter was connected to the output of the boost converter.

The experimental waveforms under different operating conditions were captured using the Chroma dedicated photovoltaic power generation monitoring software (F/W Version: Chroma ATE 61250H-600S,00368,01.10). The experimental results for the temperature change and radiation constant operation are shown in Figure 14. The radiation was 1000 W/m$^2$ and temperature was 52 °C. The MPPT

P–V curve is shown as a brightly colored thick line in Figure 14a. The MPP was 1636.50 W, then as the temperature rose to 60 °C as the light color curve shows, when downloading this I–V input file to the 6215H-600S, the experimental waveform becomes the P–V curves in Figure 14b, and the MPP was 1582.30 W.

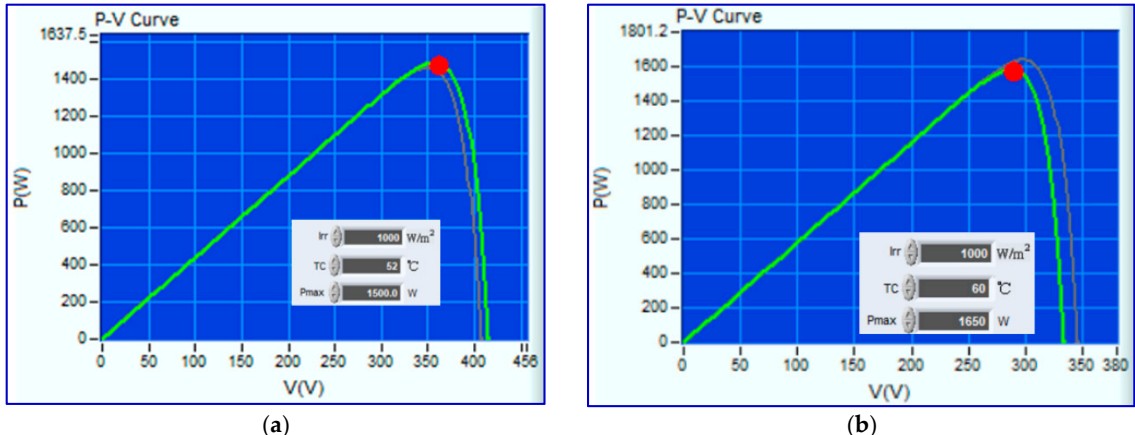

**Figure 14.** Experimental P–V tracking curve results of the constant irradiation and change temperature.
(**a**) 1000 W/m$^2$, 52 °C (**b**) 1000 W/m$^2$, 60 °C.

The maximum power reduced can be calculated as follows:

$$\Delta P_{max} = \gamma_{max} \cdot \Delta T \cdot P_{max} \tag{14}$$

According to the set experimental temperature coefficient, $\gamma_{max}$ was −0.41%/°C and the set $P_{max}$ maximum power was 1650 W of 6215H-600S output. Because the $\Delta T$ was 8 °C, the theoretical calculation was 54.12 W. Additionally, the experimental test value was 54.20 W, which could rapidly maintain maximum power tracking.

Figure 15 shows the experimental results of the MPPT tracking process under different maximum powers (the maximum power was set at 1500 W), irradiations and temperatures. When the temperature varied from 35 °C to 45 °C and the radiation synchronization increased from 500 to 1000 W/m$^2$, the MPP also changed from 798.7 to 1498.5 W, and the control output power fast tracked the new MPP quickly, as shown in Figure 15b.

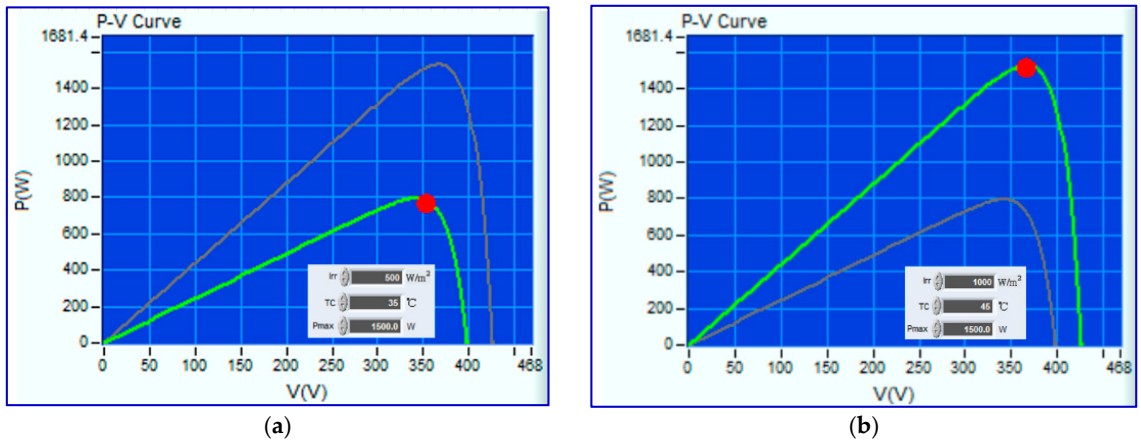

**Figure 15.** Experimental P–V tracking curve results of the change irradiation and change temperature.
(**a**) 500 W/m$^2$, 35 °C (**b**) 1000 W/m$^2$, 45 °C.

To verify the effectiveness and advantages of the proposed VUFLC-temp MPPT algorithm, comparative experiments of different MPPT control methods were conducted in the same experimental system. Figure 16 shows the experimental results of the three MPPT algorithms.

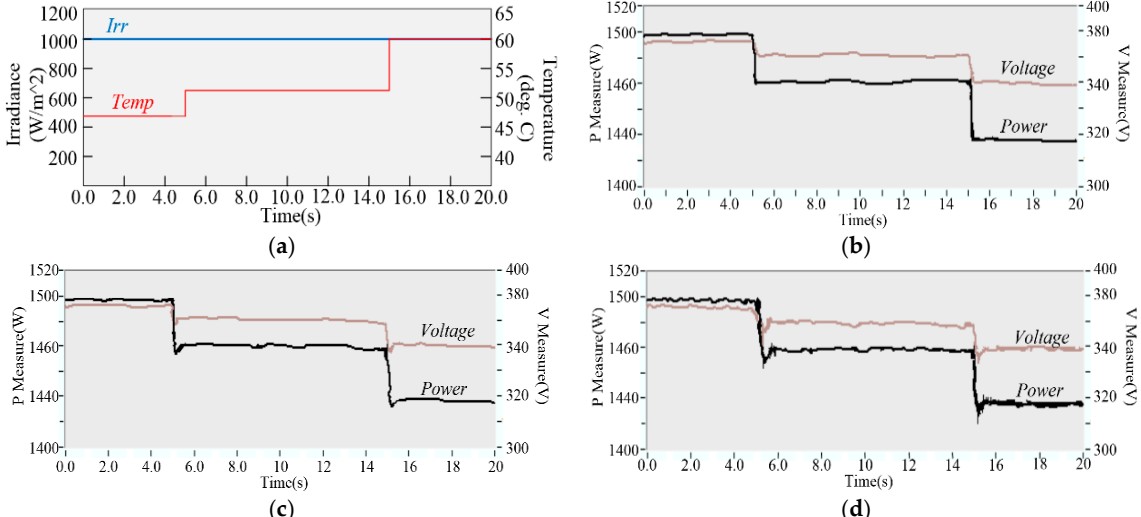

**Figure 16.** Under temperature step change: (**a**) Experimental results of MPPT control; using (**b**) VUFLC, (**c**) FLC and (**d**) INC.

As shown in Figure 16a, the irradiation intensity was a constant 1000 W/m². Before 5.0 s, the set temperature was 47 °C. At 5.0 s, the temperature steps from 47 to 52 °C. The temperature was kept at 52 °C at 10 s, then at 15.0 s, it stepped from 52 °C to 60 °C. Figure 16b–d show the results with VUFLC, FLC, and INC, respectively.

The experimental results show that the proposed VUFLC-temperature method had a faster tracking speed and smooth transition in the temperature step change in Figure 16b. However, the MPP fluctuations and oscillations occurred at temperature transition points with the FLC and INC where there were both power losses in Figure 16c,d. Furthermore, the VUFLC had better tracking stability as well as a more robust and lower static error than the others. The maximum power magnitudes and MPPT effectivenesses with different control algorithms are listed in Table 5.

**Table 5.** Experimental comparison of MPPT methods.

| Condition (1000 W/m²) | | The Experimental Results | | |
|---|---|---|---|---|
| | | INC | FLC | Proposed VUFLC |
| 47 °C | Maximum power (W) | 1481.2 | 1485.3 | 1488.9 |
| | Tracking efficiency (%) | 98.23 | 98.92 | 99.93 |
| 52 °C | Maximum power (W) | 1452.1 | 1455.8 | 1460.0 |
| | Tracking efficiency (%) | 98.01 | 98.74 | 99.90 |
| 60 °C | Maximum power (W) | 1403.9 | 1407.7 | 1411.9 |
| | Tracking efficiency (%) | 97.87 | 98.21 | 99.87 |

It can be seen that when the temperature changed, the proposed VUFLC-temperature controller had a better dynamic performance than the conventional FLC and INC control algorithms, and it was more effective at tracking and reducing the MPP oscillation.

## 6. Conclusions

In this paper, an advanced MPPT VUFLC-temperature method was proposed for a photovoltaic system, which could dynamically adjust the universe of the fuzzy controller and consider the effects of temperature changes. The output characteristics of PV cells were discussed, and according to the effects of temperature, the universe control factors were proposed and designed. Compared to the fixed universe of conventional fuzzy control, the new VUFLC-temperature MPPT method had a dynamically adjusted control factor according to the temperature change value, which could improve the MPPT tracking speed and accuracy. Different experiments were carried out. The simulation and experimental results verified the effectiveness and advantages of the proposed VUFLC-temperature MPPT method. Compared to the traditional methods, the proposed controller had a better tracking control performance under environmental changes in photovoltaic power generation systems, especially with temperature variations. The experimental results of the control system are basically consistent with the theoretical calculations when the temperature condition changes. As shown in Figure 14, the theoretical calculation and actual error is only 0.08 W. There is almost no power loss and control overshoot in Figure 16. It has the largest power generation when the temperature changes in Table 5, which is about 4 W higher than other control methods. The proposed control method not only improves the MPP tracking speed, it has the fastest tracking speed in all comparison control algorithms under the same simulation and experimental conditions, but also has higher tracking efficiency, which can improve tracking efficiency by approximately 1%.

**Author Contributions:** Y.W. proposed the main idea, designed the control system, performed the experiments and wrote the paper. Y.Y., G.F and B.Z. contributed to the discussion of this research. H.W., H.T., L.F. and X.C. double-checked and revised the whole manuscript.

**Funding:** This work was supported in part by the project of the Jiangsu Overseas Research and Training Program for the University, Science and Technology Project of Jiangsu Province Construction System, Science and Technology Planning Project of Suzhou City, and the Qinlan Project of Suzhou Vocational University.

**Conflicts of Interest:** The authors declare no conflict of interest.

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
