# Peer review of "An Advanced Maximum Power Point Tracking Method for Photovoltaic Systems by Using Variable Universe Fuzzy Logic Control Considering Temperature Variability"

_electronics, doi:10.3390/electronics7120355_

Round 1
Reviewer 1 Report
Authors propose a fuzzy-logic based approach for the solution of the mppt problem.
Paper is well written, structure is good and English quality is acceptable.
The approach is proposed in matlab and then validated with experimental equipment.
I dont have major issues with the paper, but I believe the comparison would be more fair against other fuzzy-logic based techniques cited in literature, or other soft computing techniques such as ANN based mppt.
I understand that implementing other people algorithms is very difficult and I'm not asking authors to do so, however, a table with a comparative analysis on the average mppt% and the referenced article could help assess the novelty of the paper with respect to the state of the art.
Lastly, I would expand the discussion (maybe towards conclusions) on two topics:
A) How can this approach be extended to different PV devices with different electrical characteristics (citing model identification procedures)
B) Feasibility for this Mppt approach on complex connections (series/parallel) with possibly partially shaded conditions.
Minor remarks:
In the one diode model figure, authors use "Rp" for the shunt resistance, whereas in the formulas, they use Rsh.
Eqs(3,4,5) are usually expressed with reference to the SRC conditions, not NOCT conditions.
Due to its importance, Fig.6 should be larger and with a summary description in the caption.\
Author Response
Dear reviewer,
Thank you very much for your review comments and suggestions.
About your questions,our replies are as follows:
Q_A): How can this approach be extended to different PV devices with different electrical characteristics (citing model identification procedures)
Answer:Because most photovoltaic devices or similar power generation systems have temperature characteristics, according to the control ideas proposed in this paper, it can be easily extended to other similar systems.
Q_B): Feasibility for this Mppt approach on complex connections (series/parallel) with possibly partially shaded conditions.
Answer:Because shadow occlusion is a case of photovoltaic system operation,this method can also be applied to this situation.
Furthermore, the Figure1 was corrected and modified.Added the description of Figure 6(blue color).
About the Eqs(3,4,5) ,because the PV module manufacturers often give data parameters under NCOT condition, it is more convenient to use these data for calculation.
Thanks a lot.
Best regards.
Ssincerely yours,Wang

Reviewer 2 Report
The manuscript is dealing with the variable universe fuzzy logic control (VUFLC) method applied to tracking the maximum power point (MPP) of PV system. Proposed method includes both: solar radiation intensity and temperature dependency on power output of PV system. Presented modeling and experimental results show that proposed method works better than other methods in varying weather conditions. For this reason method is interesting and paper could be published. However, modifications and improvements are necessary including following comments:
1. Line 43: “recent years [6,7]” – this references are from 2013 – use newer papers or change this sentence.
2. Fig. 1: Use Rsh as states in the text instead of Rp in the Figure.
3. Line 97: “and ….. is the reverse” – something is missing here.
4. Line 99: A is the diode ideality factor. Its value depends on the technology of the PV module ( between 1 – 2 ).
5. Line 102: “power versus voltage (Ppv-Ipv)” – it should be (Ppv-Vpv).
6. Fig. 2: use a) Different solar radiation level effect; b) change “effec” to “effect”.
7. Lines 110, 112 and others: add space between “Figure” and number, for example in line 110 is “Figure2(a)” – it should be Figure 2(a). Check and correct in the whole text.
8. Lines 111-112: “maximum electric power Pmax increases faster than the radiation” – it does not make sense. Correct this sentence.
9. Lines 113, 121 and others: use italic for Isc, Voc, Tc etc. Check and correct in the whole text.
10. Fig. 3.: This figure needs explanation. What does it mean pu? Output power of PV under STC ? How to compare temperature with power: “Temperature/pu”? Maybe, should it be Temperature / Temperature under STC (Standard Test Conditions) ? Verify and explain this in details.
11. Line 129: change “Equation” to Equations.
12. Line 128: change “coefficient” to coefficients.
13. Line 194: “The ordinary PV module” – description of the module used would be better here including table with the manufacturer data (Table 3).
14. Lines 218-219: use brackets: (-40, +40) instead of [-40, +40]
15. Line 294: Change “Table III” to Table 3.
16. Line 296: space between Table and text is missing
17. Lines 302 – 304: “In the simulation model system… operation” – this sentence should be rewritten
18. Fig. 10, Line 316: “(a) Change solar radiation and keep the temperature constant” – it is not this case, temperature is changing and irradiance is constant.
19. Lines 376 – 378: Something is wrong here including calculations. First of all the temperature coefficient of power is -0.41%/°C. From line 296 one can find out that Pmax of the system is 1650 W (under STC) instead of 1500 W. From eq. (14) you can calculate power output of the PV system for these two temperature levels: 52°C and 60°C in which temperature difference is between temperature in STC and above values. Verify theoretical calculations.
20. Line 401: Table 5 instead of Table 1 should be used.
21. Numerical results should be added to the conclusions.
22. Following sentence should be rewritten "The output mathematically expression is expressed" others were mentioned above
Author Response
Dear reviewer,
Thank you very much for your letter and the comments from the referees about our paper.
We have checked the manuscript and revised it (blue color) according to the comments. We submit here the revised manuscript.
If you have any question about this paper, please don’t hesitate to let me know.
Many thanks.
Best regards.
Sincerely yours,Wang

Round 2
Reviewer 1 Report
Authors neither gave a satisfactory answer to the comments, nor improved their manuscript accordingly.
1) The novelty has not been assessed with respect to the state of the art present in literature, as requested from the previous comments.
2) The discussion on model identification and generalization towards different devices is beyond the mere "temperature dependence" of the problem.
3) MPPT in partially shaded conditions with arbitrary series-parallel connections is not just "a case of photovoltaic system operation". Local minima appears, and generalization of MPPT techniques to these scenarios is what was asked to discuss.
Author Response
Dear reviewer:
I am very grateful to your comments for the manuscript. According with your advice, we amended and marked the relevant part in manuscript. Some of your questions were answered below.
Q1):The novelty has not been assessed with respect to the state of the art present in literature, as requested from the previous comments.
Ans:We have compared with other control techniques in Tables 4 and 5, because other control methods( such as ANN or FLC based MPPT )are relatively commonly used in the industry.There are many articles for related research.So we didn't select a specific article for comparison, so uesd well-known techniques to demonstrate the advancement and superiority of the proposed control method.Combined VFLC control with temperature influence factor was one of the most important innovations in this paper.
Q2):The discussion on model identification and generalization towards different devices is beyond the mere "temperature dependence" of the problem.
Ans:We agree with your viewpoint very much.What we want to express was that the control method proposed can be promoted in other similar cases.
Q3): MPPT in partially shaded conditions with arbitrary series-parallel connections is not just "a case of photovoltaic system operation". Local minima appears, and generalization of MPPT techniques to these scenarios is what was asked to discuss.
Ans:About this issue,we will conduct in-depth research on the application of partially shaded or other situations in next work.
If you have any question about this paper, please don’t hesitate to let me know.
Thank you very much for all your help and kind advices.I looking forward to hearing from you soon.
Best regards
Sincerely yours,Wang

Reviewer 2 Report
Authors have corrected their manuscript according to my comments in many parts. However, they didn’t give a detailed explanations to following points of the Review:
p. 10 : Fig. 3.: This figure needs explanation. What does it mean pu? Output power of PV under STC ? How to compare temperature with power: “Temperature/pu”? Maybe, should it be Temperature / Temperature under STC (Standard Test Conditions) ? Verify and explain this in details.
This part should be commented because there is no clear explanation what is pu and how both: the solar irradiance and temperature can be divided by this pu ?
p. 19. Lines 376 – 378: Something is wrong here including calculations. First of all the temperature coefficient of power is -0.41%/°C. From line 296 one can find out that Pmax of the system is 1650 W (under STC) instead of 1500 W. From eq. (14) you can calculate power output of the PV system for these two temperature levels: 52°C and 60°C in which temperature difference is between temperature in STC and above values. Verify theoretical calculations.
For power temperature coefficient wrong unit is still used. Calculations of theoretical temperatures on the basis of equation (14) should be carried out for temperatures of 52° and 60° in relation to STC (standard conditions). What is more. It should be clear from the manuscript what is real power of experimental system and why did you use different sizes of PV generator for simulation and experimental validation ? It should be the same.
p. 21. Numerical results should be added to the conclusions.
Only sentence “variation, which can improve tracking efficiency by approximately 1%” added to the conclusion is definitely not enough.
Other comments:
Line 110: “changes as the the radiation changes” – remove second “the”
Line 301: “proposed control sytem characteristics” – change to system
Lines 307 and next: instead of ° (degree sign) some other sign was used
I recommend to pay more attention on the manuscript improvement especially in abpove mentioned parts.
Author Response
Dear reviewer:
Thank you very much for your attention and comments on our manuscript.
According with your advice, we amended and marked the relevant part in manuscript. Some of your questions were answered below.
Q1):p. 10 : Fig. 3.: This figure needs explanation. What does it mean pu? Output power of PV under STC ? How to compare temperature with power: “Temperature/pu”? Maybe, should it be Temperature / Temperature under STC (Standard Test Conditions) ? Verify and explain this in details.
This part should be commented because there is no clear explanation what is pu and how both: the solar irradiance and temperature can be divided by this pu ?
Ans:The pu in the Fig.3 means “per unit”,it is the relative unit of the comparison value with the given value.We modified the annotation in the Fig.3.
Q2): Lines 376 – 378: Something is wrong here including calculations. First of all the temperature coefficient of power is -0.41%/°C. From line 296 one can find out that Pmax of the system is 1650 W (under STC) instead of 1500 W. From eq. (14) you can calculate power output of the PV system for these two temperature levels: 52°C and 60°C in which temperature difference is between temperature in STC and above values. Verify theoretical calculations.
Ans:Because the line 296 is the “Simulation PV module parameters”(the temperature coefficient of power is -0.41%/°C) ,which used the standard component library in the MATLAb software,and the 1650W is Peak Power of PV modules(In most cases it is not the actual working maximum power).We re-tested the experiment based on the new parameters and modified accordingly eq. (14)(red color in the manuscript ).
Q3): Numerical results should be added to the conclusions.
Ans:We modified and supplemented the relevant contents(red color).
Q4):
Line 110: “changes as the the radiation changes” – remove second “the”
Line 301: “proposed control sytem characteristics” – change to system
Lines 307 and next: instead of ° (degree sign) some other sign was used
Ans:We have checked the manuscript and revised it according to the comments
If you have any question about this paper, please don’t hesitate to let me know.
Thank you very much for all your help and kind advices.I looking forward to hearing from you soon.
Best regards
Sincerely yours,Wang

Round 3
Reviewer 1 Report
Authors amended to all requests.
Author Response
Dear reviewer:
Thank you very much for your suggestions . According with your advice, we carefully checked and edited the English language and style.In addition, we also applied for the English Editing Coordinator of MDPI English Editing Team to help edit the manuscript.
If you have any question about this paper, please don’t hesitate to let me know.
Thank you very much for all your help and kind advices.I looking forward to hearing from you soon.
Best regards
Sincerely yours,Wang

Reviewer 2 Report
Authors discussed my comments in answers for review and the manuscript has been updated. Some errors should still be corrected. Pay attention on grammar of English in corrected parts of the paper. Some typo errors to be removed:
Line 108: "Figure 2(a) are the curves.." - "presents the curves" sounds better,
Line 119: Isc, Voc should be italic,
123: Tc - italic,
212-214: why PS is missing ? - verify NS (positive small),
215 - 244: x1, x2, ... - italic,
263: Pmax, Voc - italic,
373: should be -0.41%/°C ,
374: Pmax - italic,
374: use ° sign after value,
378: "maximum",
372: "because",
416 - 426; font size should be the same in whole text, Use space between dots or commas and text.
418: use space between 0.08 and W,
420: space between 4 and W.
Author Response
Dear reviewer:
Thank you very much for your comments and suggestions. According with your advice, we amended and marked the relevant part in manuscript.
We carefully checked and edited the English language and style.In addition, we we also applied for the English Editing Coordinator of MDPI English Editing Team to help edit the manuscript.
If you have any question about this paper, please don’t hesitate to let me know.
Thank you very much for all your help and kind advices.I looking forward to hearing from you soon.
Best regards
Sincerely yours,Wang
